# Improving the Diagnosis of Skin Biopsies Using Tissue Segmentation

**DOI:** 10.3390/diagnostics12071713

**Published:** 2022-07-14

**Authors:** Shima Nofallah, Beibin Li, Mojgan Mokhtari, Wenjun Wu, Stevan Knezevich, Caitlin J. May, Oliver H. Chang, Joann G. Elmore, Linda G. Shapiro

**Affiliations:** 1Department of Electrical and Computer Engineering, University of Washington, Seattle, WA 98195, USA; shapiro@cs.washington.edu; 2Paul G. Allen School of Computer Science and Engineering, University of Washington, Seattle, WA 98195, USA; beibin@uw.edu; 3Pathology Department, Isfahan University of Medical Sciences, Isfahan 8174673461, Iran; mokhtari.ptlgy@gmail.com; 4Department of Biomedical Informatics and Medical Education, University of Washington, Seattle, WA 98195, USA; wenjunw@uw.edu; 5Pathology Associates, Clovis, CA 983611, USA; stevanrk@gmail.com; 6Dermatopathology Northwest, Bellevue, WA 98005, USA; campbell.cait@gmail.com; 7Department of Pathology, University of Washington, Seattle, WA 98195, USA; ochang@uw.edu; 8David Geffen School of Medicine, UCLA, Los Angeles, CA 90024, USA; jelmore@mednet.ucla.edu

**Keywords:** whole slide imaging, skin biopsy, melanoma diagnosis, machine learning, semantic segmentation, transformers, accuracy

## Abstract

Invasive melanoma, a common type of skin cancer, is considered one of the deadliest. Pathologists routinely evaluate melanocytic lesions to determine the amount of atypia, and if the lesion represents an invasive melanoma, its stage. However, due to the complicated nature of these assessments, inter- and intra-observer variability among pathologists in their interpretation are very common. Machine-learning techniques have shown impressive and robust performance on various tasks including healthcare. In this work, we study the potential of including semantic segmentation of clinically important tissue structure in improving the diagnosis of skin biopsy images. Our experimental results show a 6% improvement in F-score when using whole slide images along with epidermal nests and cancerous dermal nest segmentation masks compared to using whole-slide images alone in training and testing the diagnosis pipeline.

## 1. Introduction

Melanoma is one of the deadliest types of skin cancer, and its incidence has been increasing faster than any other cancer [1,2,3]. If Melanoma is caught in its earlier stages, it is highly curable; however, because of the complexity of skin biopsies and the subjectivity of visual interpretation, there is significant uncertainty in the accuracy of pathology reports. Studies have shown that pathologists’ diagnoses of moderately dysplastic nevi to thin invasive melanomas are neither accurate nor reproducible in some cases [4]. These reports raise concerns about appropriate treatment and the consequences of both under- and over-diagnosis. Deep learning has shown excellent performance on various tasks, and healthcare is not an exception [5,6,7,8]. Using deep-learning techniques to provide prognostic and diagnostic information for pathologists during screening and treatment stages can be an aid in clinical care.

Deep learning and artificial intelligence (AI) have achieved unparalleled success in various tasks such as classification, segmentation, detection, etc. However, though the state-of-the-art approaches in this field show fast and accurate performance, they face challenges in dealing with medical datasets. Medical datasets usually are small in sample size, have large images, and do not have many examples of perfect annotations. As the field of AI in healthcare has grown significantly in recent years, more robust methods in this area have emerged.

In addition, demand for diagnostic models and classification tools based on histopathological images has increased due to inter- and intra-observer variability in pathology and the potential solution that AI methods can produce. Providing prognostic and diagnostic information at the time of cancer diagnosis has important implications on patient outcomes, as automated machine-learning methods on whole-slide images provide a promising way forward for efficient and robust pathology analysis.

Various studies have introduced diagnosis models based on whole slide images (WSIs). In [9], the authors introduced a CNN-based deep feature extraction framework to build slide-level feature representations via weighted aggregation of the patch representations and overcome the challenge of working with variable-sized regions of interest. Li et al. [10] extracted relevant patch representation using self-supervised contrastive learning and introduced a dual-stream architecture with trainable distance measurement to train an MIL model called the dual-stream multiple instance learning network (DSMIL). Chikontwe et al. [11] proposed a multiple instance learning (MIL) method based on a transformer that first selects the top-k patches, and then used these patches for instance-learning and bag-representation learning. In addition, this method uses a center loss that maps embeddings of instances from the same bag to a single centroid and reduces intra-class variations for final diagnosis.

Segmentation-based methods are another approach that has been studied in the field of histopathology image analysis, as different tissues and entities in these images might play an important role in the diagnosis of the case. Several works with this approach first generate semantic segmentation masks on WSIs, and using the extracted information from those masks, produce an image-level diagnosis [12,13,14]. While this approach is a valuable study direction, the challenge of dealing with imperfect annotation or lack of annotation is not addressed in such studies.

In our prior AI-based diagnosis work in pathology, our studies utilize regions of interest (ROI) rather than the larger whole slide images (WSI) [9,13,15]. There are two main reasons we used the full WSI for the current study. First, Mercan et al. [16], in the effort to find diagnostically relevant ROIs on breast biopsy WSI, reported that 74% of the output probability map overlapped with the actual ROIs from pathologist viewing behavior, while 26% did not. If such early probability maps are utilized for diagnosis tasks, there is a chance that important diagnostic information is missed or misused. The second reason behind our approach using WSI relates to the interpretive process used by pathologists as they view, assess, and interpret WSI of skin biopsies using current published definitions for clinical classification. The pathologists’ clinical process and classification systems vary by tissue type—for breast biopsy cases, a single ROI of an area within a duct might suffice to allow the pathologists to come to a diagnosis. However, the process used by pathologists of reviewing skin biopsy image data and the information within skin biopsies used to determine a diagnosis is different—information on the image from larger structural data in addition to image data within small clusters of cells is important to both rule in and rule out different diagnoses. Thus, for a diagnosis of melanoma and its precursors, reviewing information from the larger WSI is required in current clinical practice by pathologists before they can provide a diagnosis.

In this work, we therefore incorporate tissue segmentation masks that were generated based on sparse and coarse annotations of the full skin biopsy WSIs. The goal is to investigate the potential of providing this information in the process of skin biopsy diagnosis using WSI. Our experimental results show that including a clinically certain important tissue structure along with WSIs improves the learning of the model, especially in challenging diagnostic classes such as melanoma in situ (MIS) and invasive melanoma (T1a). Examples of tissue structures that show the highest improvements are Epidermal Nests and melanoma dermal nests (cancerous). These tissues are considered clinically important in the decision-making process by human pathologists. Comparing our results with 187 pathologists’ performance on the same test set shows that our model can outperform or have comparable performance on the cases with the aforementioned diagnostic classes.

## 2. Materials and Methods

### 2.1. M-Path Dataset

Our dataset comes from the M-Path study [4] that was approved by the Institutional Review Board at the University of Washington (protocol number STUDY00008506) and was conducted by a Bellevue, Washington dermatopathology laboratory. Two-hundred-and-forty hematoxylin and eosin (H&E)-stained slides of digitized skin biopsy images from this study are included in our project and can be classified into five different MPATH-Dx (melanocytic pathology assessment tool and hierarchy for diagnosis) simplified categories based on presumed risk of the lesion and suggested treatment recommendations [17]. Example diagnostic terms for each MPATH-Dx class are as follows: (I) mildly dysplastic nevi, (II) moderately dysplastic nevi, (III) melanoma in situ and severely dysplastic nevi, (IV) invasive melanoma stage T1a, and (V) invasive melanoma stage ≥ T1b. Table 1 shows the distribution of the diagnostic categories of the M-path dataset. Figure 1 shows examples of three different WSIs in the M-Path dataset.

Using the MPATH-Dx classification tool [18] that is described above, a consensus panel of three dermatopathologists with internationally recognized expertise made a consensus diagnosis for all cases. Following these meetings, the expert panel, as well as an additional dermatopathologist (S. Knezevich), assigned one rectangular area as a region of interest (ROI) per case. These ROIs represent an important area of the WSI for diagnosis. Since there was a limitation of one ROI per case, there might have been other diagnostically important regions on WSIs that are not included in the final ROI. However, assigned regions have valuable information that can be used for various purposes. These variable-sized ROIs (Figure 2) can be extracted using their coordinates.

To reduce the input image size and eliminate the unnecessary information from the slides’ orientation (since this information is not relevant to the diagnosis of a case), we used extracted slices from the WSIs. An example of a WSI and its corresponding extracted slices is shown in Figure 3.

#### 2.1.1. Segmentation Masks

In a previous study [19], using coarse and sparse annotations, we trained a two-stage segmentation pipeline that generates tissue segmentation masks on whole slide images of skin biopsies. The segmentation masks include epidermis (EP), dermis (DE), stratum corneum (COR), epidermal nests (EPN), dermal nests (DMN), and background (BG). In the first stage, using a U-Net model, a model is trained that is able to segment large entities such as dermis, epidermis, stratum corneum, and background. In the second stage, two models are trained on the smaller tissue structures of the skin biopsy images. This stage includes two branches that are trained separately: (1) stage 2-dermis, which uses a U-Net to train a model on the dermis portion of the image (i.e., DMN); (2) stage 2-epidermis, which trains a U-Net on the epidermis portion of the image (i.e., EPN).

Using this pipeline, we were able to generate segmentation masks for both large entities (i.e., dermis, epidermis) and smaller entities (dermal nests, epidermal nests) with high-quality performance. However, since the annotations of DMN and EPN were coarse, we observed over-labeling of these entities in segmentation results as well. Figure 4 shows some examples of segmentation masks generated from WSIs.

### 2.2. Dermal Nest Classification

Pathologists investigate structural entities in digitized whole-slide images of melanocytic skin lesions and assign a diagnosis class to the case based on the various factors, including morphological characteristics of the cells present in the biopsy images. Assessment of the architecture and cytomorphology of junctional (epidermal) melanocytes and dermal melanocytes is necessary to classify and risk-stratify melanocytic lesions.

The evaluation of dermal nests is key in distinguishing a melanocytic nevus from invasive melanoma. It can also represent one of the most challenging tasks for a pathologist, especially in the absence of additional lab testing. Generally, dermal nests are categorized into the two sub-groups of nevus nests and melanoma nests. Dysplastic melanocytic nevi and severely dysplastic nevi may contain benign dermal melanocytic nests, but only invasive melanoma contains malignant dermal melanocytic nests.

In [19], we proposed a two-stage segmentation pipeline in which epidermal nests (EPN) and dermal nests (DMN) were segmented in its second stage. However, since not enough examples of nevus dermal nests (DMN-N) were available, especially compared to other entities such as melanoma dermal nests (DMN-M) and epidermal nests (EPN), we decided to combine nevus dermal nests (DMN-N) and melanoma dermal nests (DMN-M) into one class of dermal nests (DMN) in that project. In this paper, we propose an additional step to the output of our segmentation model that allows us to classify segmented DMNs into two sub-categories of nevus or melanoma. We train a CNN model that is able to segment dermal nest into melanoma dermal nest and nevus dermal nest. The classes of epidermal nests, melanoma dermal nests, and nevus dermal Nests can now be used in our experimental pipeline.

#### 2.2.1. Dermal Nest Dataset

To train a dermal nest classifier, some ground truth on different categories of dermal nests is required. The ground-truth annotations in this project are a subset of the coarse and sparse annotations that were introduced in Section 2.1.1. The original set contained a small number of examples of nevus dermal nests on ROIs with the diagnostic classes of I, II, and III, while there was a relatively larger number of examples of melanoma dermal nests on ROIs belonging to cases with diagnostic classes IV and V. The main challenges in working with these annotations were two-fold: (1) There was a huge gap between the sample size of nevus dermal nests and melanoma dermal nests in which melanoma dermal nests contained ~400 M pixels, which is eight times the size of nevus dermal nests with ~50 M pixels and (2) no examples of nevus dermal nests on any invasive melanoma cases were annotated. The only examples of dermal Nest annotation in these classes belonged to melanoma dermal nests, while in reality, both types of nests can be present in one invasive melanoma case. Hence, in the segmentation model of [19], all dermal nest annotations were combined into a single class of dermal nests (DMN). Figure 5 shows example annotations of nevus dermal nests (DMN-M) (Figure 5b) and melanoma dermal nests (DMN-M) (Figure 5e) and their conversion to dermal nests (DMN) ((Figure 5c) and (Figure 5f)), which were used for the dataset of [19].

In this paper, instead of combining the two types of dermal nests, we kept them separate and extracted them into two categories of nevus dermal nests (DMN-M) (Figure 5b) and melanoma dermal nests (DMN-M) (Figure 5e). For the nest extraction step, after masking out everything other than dermal nests in the ROIs, we sampled the nests into two classes of “nevus” and “melanoma”. The sampling window size is 100 × 100. As expected, there was a noticeable imbalance in the final dataset between the two classes of “nevus” and “melanoma” nests. The number of extracted nevus nests was 604 samples, while the number of extracted melanoma nests was 5732 samples. To solve this imbalanced dataset issue, we used the result of our previous segmentation model as explained in Section 2.2.2.

#### 2.2.2. Solving Nest Sample Imbalance in the Training Dataset

After acquiring the segmentation model output, the opportunity of overcoming the annotation imbalance in dermal nests arises. It is known that cases with a diagnosis class of I, II, and III only contain nevus dermal nests, while both nevus dermal nests and melanoma dermal nests can appear in a case with diagnostic class IV or V. Although the segmentation model of [19] does not distinguish between nevus dermal nests and melanoma dermal nests, we know that all the nests on class I, class II, and class III cases are nevus dermal nests. The reason is that if there is any appearance of a melanoma dermal nest on a case, that case will move to one of the invasive melanoma diagnostic categories. Figure 6a shows an example of segmented dermal nests on a class II case in which we assume all nests are of nevus type based on the diagnosis of the case. Figure 6b shows an example of segmented dermal nests on a class V case. Such a case would not be usable for training in this project since it is not specified which parts of the segmented dermal nests are nevus and which parts are melanoma.

Since the training and testing split of the dataset is consistent throughout all the projects, in addition to the fact that all the dermal nests in cases with diagnosis classes of I, II, and III must be nevus dermal nests, it is only logical to apply the trained segmentation model from [19] to WSI of cases I, II, and III to detect dermal nests (DMN); extract them; and re-label them as nevus dermal nests. Using the new nevus dermal nests, we can randomly extract DMN-N samples and add them to the nest classification training set to reach a balanced number of samples for both classes of DMN-N and DMN-M in the training set.

#### 2.2.3. Dermal Nest Classifier

Since convolutional neural networks (CNNs) have shown good performance in various computer vision and machine-learning tasks, we used this approach in training our dermal nest classifier. We trained three different architectures, using PyTorch torchvision [20] pre-trained CNN models, trained on the ImageNet dataset [21]:DenseNet: Huang, G., et al. [22], introduced a densely connected convolutional neural network that improves the flow of information between different stacked convolutional layers. In our experiments, we used a pre-trained torchvision *densenet161* architecture as a nest classifier model.ShuffleNet: ShuffleNet [23] is a convolutional neural network that utilized two new operations, point-wise group convolution, and channel shuffle, to reduce computation cost while maintaining accuracy. We used a pre-trained torchvison *shufflenet_v2* for our experiments.ResNet: A residual neural network [24] is a CNN that utilizes skip connections to jump over some layers. We used a pre-trained torchvison *resnet18* for two of our experiments with different training datasets.

In the preprocessing step, we included random cropping, random rotation, horizontal flip, and normalization in the Dataloader function. All the models were trained for 20 epochs with cross-entropy [25] as a loss function, and Adam optimizer [26] with a learning rate of 0.001. After the training, we evaluated each model’s performance on the same testing dataset and compared the results.

### 2.3. WSI Diagnosis Using Tissue Segmentation

In this section, we study the impact of adding each tissue mask to the WSIs in the classification of our dataset into diagnostic categories. The M-path dataset described in Section 2.1 with five diagnostic classes of (1) Class I: mild dysplastic nevi, (2) class II: moderate dysplastic nevi, (3) Class III: (e.g., melanoma in situ and severely dysplastic nevi), (4) Class IV: invasive melanoma stage T1a, and (5) Class V: invasive melanoma stage ≥ T1b. The only difference is that since the clinical risk for progression of both Class I and Class II is extremely low, and we have a limited sample size in the aforementioned classes, we regrouped the five classes to four diagnostic classes by combining samples from class I and II into one class. The final four classes will be (1) Class I–II: mild and moderate dysplastic nevi (MMD), (2) Class III: (e.g., melanoma in situ, severely dysplastic nevi) (MIS), (3) Class IV: invasive melanoma stage T1a (T1a), and (4) Class V: invasive melanoma stage ≥ T1b (T1b).

As mentioned in Section 2.1, we used extracted slices to train and evaluate our diagnosis models. The main resolution that we used to extract individual slices was 20×. Using this resolution, we extracted lower resolutions of 7.5×, 10×, and 12.5×, which we later used for our experimental studies.

#### 2.3.1. Binarized Segmentation Masks

The segmentation masks generated by the proposed pipeline in Section 2.1.1 were used in the current project. Each tissue mask from that project (epidermis (EP), dermis (DE), epidermal nest (EPN), and dermal nest (DMN)) was separated into a single binary mask in order to have more control over tissue combination in our experimental studies on the diagnosis accuracy. In addition to the aforementioned tissue masks, we included the two types of dermal masks from 2.2 as two separate binary masks of nevus dermal nest (DMN-N) and melanoma dermal nest (DMN-M). Figure 7 shows examples of binary masks for two classes of mild and moderate nevi (MMD) and invasive melanoma stage ≥ T1b (T1b). Note that the moderate nevi (MMD) case does not include any DMN-M; hence, the corresponding mask is all zeros.

#### 2.3.2. Dataset Split

The dataset of WSIs before the extraction of slices was divided in half, conserving the original set’s diagnostic class distribution over both subsets. One-half of the dataset was used for training and validation subsets, and the other half of the dataset was kept unseen from the model during the training and solely used for the final evaluation of the trained model. This split was kept fixed over all the experiments. After splitting the dataset, the extraction step that is explained in Section 2.1 was applied to all the WSIs in the training, validation, and testing subsets.

#### 2.3.3. Soft Labels

Usually, each WSI has multiple slices from the same skin biopsy; however, not all the slices contain related information to the assigned diagnostic class of the case. In clinical practice, if a pathologist detects invasive melanoma in just one or two slices on one case, the overall biopsy is diagnosed as invasive melanoma to guide clinical care and treatment. In our dataset, the ROIs (some examples in Figure 2) that helped pathologists in diagnosis belong to one or two tissue slices, while the other tissue slices may correspond to other diagnostic categories. If all the extracted slices from a WSI are assigned to one diagnostic class, there is the risk of false representation of that diagnostic class, which can interfere with the learning process of a model. To handle this issue, we used a method that was previously developed by our group in which, using a singular-value decomposition (SVD), soft labels are assigned to the slices that do not have an ROI on them. For more information about the details of this method, refer to [27].

#### 2.3.4. Combining WSI and Segmentation Masks

We tried various methods to combine the information from WSIs and corresponding segmentation masks. The final method that we chose to implement and run our experiments is as follows: Each WSI has three channels of RGB: red (R), green (G), and blue (B). In order to add segmentation mask information to our data, we concatenate each mask as a new channel to the image. For example, if we add a DMN channel to the WSI, we will have a new input with four channels: R, G, B, and DMN. This approach gives the flexibility of investigating any combination of tissue masks that are of interest. In addition, the feature extractor obtains the information of appended tissue masks along with the original WSI, which might result in a more representative feature set.

#### 2.3.5. Feature Extraction

We used MobileNetv2 [28] pre-trained on the ImageNet dataset [21] as a feature extractor on our extracted patches. MobileNetv2 outputs 1280-dimensional patch-wise features after global average pooling. Since the pre-trained network on the ImageNet dataset is essentially a network with three input channels of RGB, we modified the first layer of the network by replacing it with a *Conv2d* layer that has input channels equal to the number of input image channels. The number is not fixed since, as explained in Section 2.3.4, the number of input image channels depends on the tissue mask combination in a specific experiment. Changing the first layer of the network, which is not pre-trained on any image, has the potential of negatively impacting the feature extraction step; however, as we will see in the next sections, the results do not show any clear effect of such. The reason might be the nature of CNNs in which the first few layers are focused on low-level features, while the middle layers mainly extract high-level and fine detailed features.

#### 2.3.6. Scale-Aware Transformer Network (ScATNet)

In previous work, Wu et al. [27] proposed scale-aware transformer network (ScATNet) for diagnosing melanocytic lesions using WSIs. ScATNet uses local and global representations from various scales. In this architecture, the first step is to learn local patch-level embeddings on each scale using a pre-trained CNN. Then, using a transformer, the model learns the contextualized patch embeddings for each scale. In the last step, scale-aware embeddings across various scales are trained to the model [27].

ScATNet projects extracted patch-wise features explained in Section 2.3.5 linearly to a 128-dimensional space. In the second and third steps of the ScATNet pipeline, a stack of two transformer units is used. Each transformer unit has four heads in the self-attention layer with a feed-forward dimension of 512.

#### 2.3.7. Experimental Studies

In order to investigate the impact of different tissue types, we designed several experiments with various combinations of tissue segmentation masks, using ScATNet as the basic model. In each experiment, we included specific segmentation masks along with the WSI; extracted the features as explained in Section 2.3.5; and using the extracted features, we trained and tested a diagnosis model. We ran the experiments with various resolution scales (7.5×, 10×, 12.5×, combination of two scales, and all three scales), with different hyperparameters, and after finding the best setting, we ran all the experiments with different random seeds.

Figure 8 shows an overview of our approach.

#### 2.3.8. Hyperparameters

ScATNet was trained for 200 epochs in an end-to-end fashion using the ADAM optimizer with a linear learning rate warm-up strategy and step learning rate decay. The best result in our experimental studies was achieved using a single scale of 7.5×.

## 3. Results

### 3.1. Dermal Nest Classification Results

All the models from both approaches were evaluated by a testing set of ROI images that was kept unseen from the model during the training process. Note that in the testing dataset, no nest samples from the segmentation model are included. The testing dataset only contains extracted nests from ROIs in which we had a pathologist’s annotation as ground-truth to compare model prediction against them. Using the model with the best performance on ROI images, we generate DMN-M and DMN-N on extracted slices of the WSI.

#### 3.1.1. Quantitative Results on ROIs

All the trained models were evaluated on the same ROI testing set. Each nest classifier’s performance was measured using these metrics: F-score, precision, sensitivity (recall), and specificity. The results of this evaluation are summarized in Table 2.

As a model selection step after training each experiment for 200 epochs, and to improve the model’s robustness against stochastic noise, we averaged the best five model checkpoints within a single training process inspired by [29]. Then we evaluated all our experiments over the same testing set. A WSI might contain multiple tissue slices, which were extracted into single slices, and each of these slices might have a different diagnostic class prediction. To decide on the final diagnosis of a specific WSI, we used max-voting, which means if one of the tissue slices in a WSI is invasive melanoma, then the entire WSI corresponds to invasive melanoma and cannot be MMD or MIS. This approach was inspired by how pathologists make their diagnosis decision on skin biopsy images.

#### 3.1.2. Qualitative Results on WSIs

After acquiring our best nest classifier (ResNet), we ran the model on all the dermal nests (DMN) extracted from the previous segmentation mask of invasive melanoma stage T1a and ≥ T1b WSIs to generate melanoma dermal nests (DMN-M). Any segmented DMN samples in these classes that were not classified as a DMN-M by the nest classifier model are assigned to nevus dermal nest (DMN-N). Figure 9 shows examples of an extracted slice of invasive melanoma WSI, corresponding dermal nest mask generated by our previous segmentation model, melanoma dermal nest (DMN-M) portion of the dermal nest (DMN) as a result of nest classifier output, and nevus dermal nest (DMN-N) portion of dermal nest (DMN) as a result of the complement of DMN-M on DMN.

### 3.2. Diagnosis Experiment Results

We evaluated all the models based on micro F-score, sensitivity (recall), and specificity. Note that in dealing with a multi-class classification, where every test datum should belong to only 1 class and not multi-label, we cannot use the same F-score as in binary class classification (i.e., macro F-score in multi-class classification). The correct way to report an F-score in multi-class classification is to calculate the micro-averaged F-score (AKA micro F-score) based on micro-precision and micro-recall. Micro-precision measures the precision of the aggregated contributions of all classes, and micro-recall measures the recall of the aggregated contributions of all classes.

Micro_precision = TP_sum_/(TP_sum_ + FP_sum_)Micro_recall = TP_sum_/(TP_sum_ + FN_sum_)Micro F-score = 2 × (micro_precision × micro_recall)/(micro_precision + micro_recall)Sensitivity (recall) = TP_sum_/(TP_sum_ + FN_sum_)Specificity = TN_sum_/(TN_sum_ + FP_sum_)

#### 3.2.1. Experimental Results

The summary of the results is shown in Table 3. The F-score of each experiment is reported based on 10 different random seeds, along with average sensitivity and specificity over the 10 random seeds per experiment. In our experiments, the (average, max) F-scores were (0.54, 0.58) for the raw WSI with no segmentation masks, which improved to a high of (0.60, 0.62) for the raw WSI plus the epidermis mask and the dermal melanoma mask (i.e., the cancerous nests in the dermis). The addition of the dermal melanoma mask was important as it gave a significant gain over just providing dermal nests. Note that we started with a rather low F-score for the raw WSI and fixed those parameters to achieve stability, so it is possible that even higher values [27] can be achieved by starting with a different set of parameters for the WSI run. However, we favored stability, and the (0.54, 0.58) scores were stable, in that they could be achieved repeatably.

#### 3.2.2. Comparison of Confusion Matrices

Table 4 shows a comparison of two experiments’ confusion matrices. Table 4a is an example of a multi-class confusion matrix of experiments that only contain RGB channels of the WSI in the dataset, while Table 4b shows an example of an experiment in which we had R, G, and B channels of the WSI along with two extra channels of epidermal nest (EPN) binary segmentation mask and melanoma dermal nest (DMN-M) binary segmentation mask (a total of five channels per image).

As shown in the tables, the number of true positives (TP) of classes MIS, T1a, and T1b increased in the experiment in which we included segmentation masks along with WSI. Another important finding is that the misclassified cases of MIS when we have EPN and DMN-M information are mostly on T1b. In the real world, MIS is a challenging case for pathologists to make a definite diagnosis. The comparison of confusion matrices in Table 4 and tissue experiments’ results in Table 4b shows that the model is able to learn more information when segmentation masks are introduced along with the WSI, which can be an assistance to pathologists in challenging cases.

#### 3.2.3. Single-Scale vs. Multi-Scale

In our experiments, we ran each setting of tissue experiments with single scale, two scales, and three scales. A summary of results for one example tissue experiment (WSI + EPN + DMN-M) in comparison with a raw WSI, which has the exact same parameters and scales, are summarized in Table 5. These results suggest that having segmentation masks does not improve the performance when ScATNet is trained on multiple scales, and the gain of improvement is lower when the higher resolution of WSI along with segmentation masks is used.

This behavior can be explained by the specific strategy of ScATNet in patching input images on different scales. For example, images in 7.5× resolution are divided into 5 × 5 = 25 crops while 12.5× images are divided into 9 × 9 = 81 crops. In addition, the transformer unit in the ScATNet architecture includes a self-attention module that learns to pay more attention (i.e., assign higher weight) to specific patches in an image. When we introduce a WSI along with its corresponding dermal nests and epidermal nests, the model learns during the training process that these structures are important in decision making. Hence, when these tissue structures appear in a testing case’s segmentation mask, the model assigns higher weights to the patches that contain those structures. If a segmentation mask of a testing case is inaccurate, especially when some important structures are over-labeled, it can negatively impact the model’s decision-making and lead to a false prediction. The possibility of such an impact could be higher in higher resolutions since there will be more patches with inaccurate tissue labels; hence, higher weights on irrelevant patches. Figure 10 shows an example of a test set WSI and corresponding segmentation mask (Figure 10a) that includes dermis, epidermis, melanoma dermal nest, epidermal nest, corneum, and background. The segmentation of epidermal nests is inaccurate and over-labeled, and potentially led to a wrong prediction on resolution 12.5× (Figure 10c), since the number of patches with noise at that resolution is more than at resolution 7.5× (Figure 10b).

#### 3.2.4. Comparison to US Pathologists

We have access to the interpretation of 187 US pathologists on the same testing set that we used in our experimental studies. Table 6 shows the comparison of the F-score, sensitivity, and specificity of pathologists’ performance and our best model (WSI + EPN + DMN-M) performance. We observe that our model either outperforms the pathologists’ results on the challenging classes of MIS and T1a or has a comparable performance. This finding shows the potential that providing an assistant tool can have in the time of cancer diagnosis and treatment.

#### 3.2.5. Comparison to Other Baselines

We compared our results with several other methods developed to make a diagnosis based on histopathology images.

Weighted Feature Aggregation: Deep Feature Representations for Variable-Sized Regions of Interest was introduced by [9]. In this method, a CNN-based deep feature extraction framework builds slide-level feature representations via weighted aggregation of the patch representations. In this pipeline, the patch-wise feature will be extracted by a VGG16 pre-trained CNN, then using two different approaches of either penultimate layer features (penultimate-weighted) or hypercolumn features (hypercolumn-weighted), the features are concatenated in a weighted manner. As the last step, using average pooling, a slide-level representation is generated, which is later used for training and testing the diagnosis CNN model.Dual-stream Multiple Instance Learning Network (DSMIL): In this work, Li et al. [10] used self-supervised contrastive learning to extract good representations from patches and using an aggregator that models the relations of the instances in a dual-stream architecture with trainable distance measurement, trained a MIL model.Multiple Instance Learning with Center Embeddings (ChikonMIL): [11] proposed a multiple instance learning (MIL) method that first selects the top-k patches, and then uses these patches for instance-learning and bag-representation learning. In addition, this method uses a center loss that maps embeddings of instances from the same bag to a single centroid and reduces intra-class variations for the final diagnosis.

The results of all the baseline methods and their comparison with our best model are summarized in Table 7. Our model using the epidermal nests and dermal melanoma nests is able to beat all of them.

## 4. Discussion

The rapidly growing number of melanoma cases along with inter- and intra-observer variability of diagnosis by human pathologists is of concern in this field. On the other hand, advances in machine learning and artificial intelligence methods have presented the potential to provide assistant tools for the pathologists to analyze whole-slide images (WSIs) for diagnosis and prognosis objectives.

In recent years, interest in artificial intelligence research in various fields including healthcare has been increasing rapidly. Deep-learning methods have shown impressive and robust performance on various tasks and hold promise for providing assistant tools in healthcare research including pathology. Dermatopathology research is not an exception in benefiting from the advancement of artificial intelligence [30,31,32]. In the time of cancer monitoring and treatment, AI-developed tools have the potential to assist dermatopathologists especially with challenging cases. In addition, the educational and research aspects of AI-developed methods in tutoring practicing pathologists introduce new prospects for reducing the diagnostic errors in clinical care.

In recent years, deep-learning methods have proven to have excellent performance in different tasks such as image classification. However, most of the state-of-the-art methods either require a fairly large dataset to train a model or a large amount of pixel-level annotation. Both of these requirements are a challenge in dealing with medical datasets as these datasets are usually small, especially compared to general datasets such as ImageNet [21], and obtaining fine manual annotation on them is not a time or cost-effective task.

In this work, we proposed an approach that uses the segmentation masks that we previously obtained using sparse and coarse annotation [19], and adds information to WSI from a dataset of skin biopsy images. In this work, we first designed a dermal nest classifier that can classify segmented dermal nests (DMN) into two sub-categories of nevus dermal nests (DMN-N) and melanoma dermal nests (DMN-M). Using the previous and new masks, the goal was to investigate the potential of each important tissue mask in skin biopsy images to improve the results of a multi-class diagnosis model.

Our experiments showed that including certain segmentation masks along with WSIs yields a better diagnosis output with one scale. One of the foremost tissue types in skin biopsy images are nests that contain various types such as epidermal nests (EPN), nevus dermal nests (DMN-N), and melanoma dermal nest (DMN-M). We observed significant improvement when including EPN and DMN-M (which is considered the cancerous type of dermal nests) along with the corresponding WSI, compared to the experiments that do not include any segmentation masks. Further analysis showed that including the aforementioned entities improved the learning of the model on invasive melanoma and melanoma in situ, which are challenging classes on which to make a consensus decision. Improvement in the challenging classes proves the potential AI has in healthcare and pathology.

As mentioned in Section 2, each WSI in the M-path dataset has an expert-assigned ROI that carries either important diagnostic or prognostic information. However, since (1) experts were limited to one ROI per case, and (2) the diagnosis of some skin biopsies requires review of the full whole slide image, we designed our diagnosis pipeline to utilize full WSIs rather than a single ROI per case. One might wonder if multiple ROIs would be sufficient in place of the WSI, and perhaps more efficient. For example, if multiple ROIs were generated by an AI program for use in diagnosis, these may actually slow down the diagnosis process if provided to human pathologists who are used to their own way of examining slides. If multiple ROIs were identified by expert pathologists and were provided to a computer program classifier, it would not know which (if any) were more important, and thus, looking at the WSI is still the best course for the computer diagnosis for skin biopsy specimens.

Certain limitations need to be considered. The dataset that we used in this project is small and of melanocytic skin lesions, and while the cases included were carefully selected to represent the full spectrum of cases in clinical practices in the US, we are not certain how well the method would perform on the full spectrum of skin biopsies (e.g., including non-melanocytic lesions). In addition, the sparse annotations for the segmentation project were provided on ROIs on the WSI, which means there was prior knowledge of which part of the WSI contained valuable information. Not all medical datasets benefit from having ROI assigned to each case.

The unique strengths of this work include the ability to compare our results to the diagnostic interpretations given to the cases by actively practicing U.S. pathologists. This comparison showed that our model could outperform or have comparable performance to pathologists in some challenging classes. Ours is the first deep-learning model to add segmentation data of the clinically important tissue structure to the raw images to improve melanoma diagnosis. Since our segmentation model was trained on a sparse and coarse annotation set, providing a diagnosis pipeline that improves the outcome by leveraging the imperfect segmentation masks highlights the potential of AI approaches in dealing with challenges and shows a promising future for AI in healthcare.

## Figures and Tables

**Figure 1 diagnostics-12-01713-f001:**
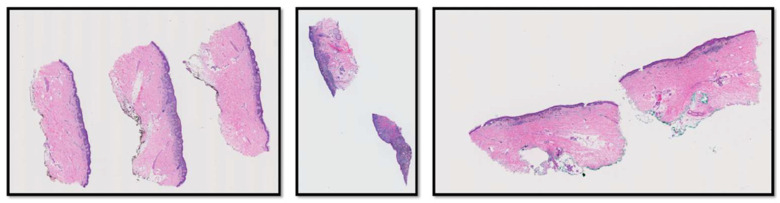
Three examples of WSIs in the M-Path dataset. The left image is a case with class IV diagnosis (invasive melanoma stage T1a), the middle image is a case with class V diagnosis (invasive melanoma stage ≥ T1b), and the right image is a case with class IV diagnosis (invasive melanoma stage T1a).

**Figure 2 diagnostics-12-01713-f002:**
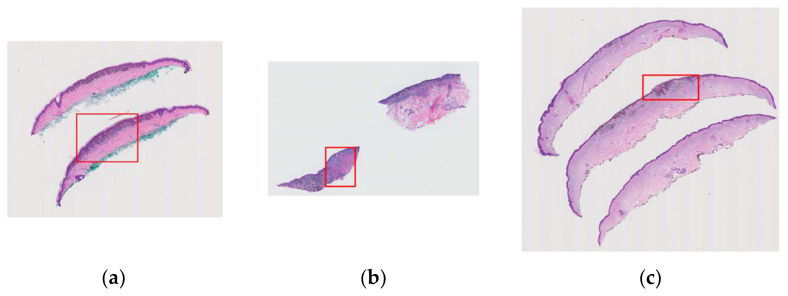
Examples of variable-sized region of interests (ROI) assigned by pathologists that contain important diagnostic information are shown in red boxes: (**a**) a case with class II diagnosis (moderately dysplastic nevus), (**b**) a case with class V diagnosis (invasive melanoma stage ≥ T1b), (**c**) a case with class IV diagnosis (invasive melanoma stage T1a).

**Figure 3 diagnostics-12-01713-f003:**
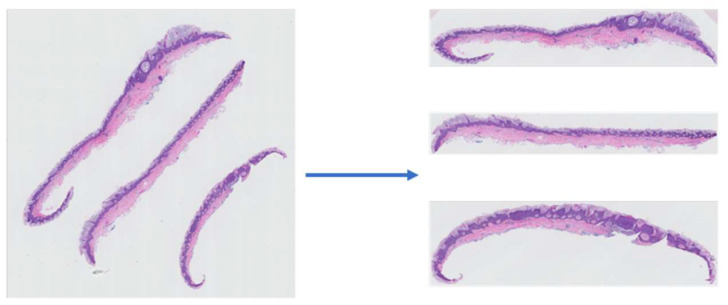
An example of a WSI (**left**) and its corresponding slice extraction (**right**).

**Figure 4 diagnostics-12-01713-f004:**
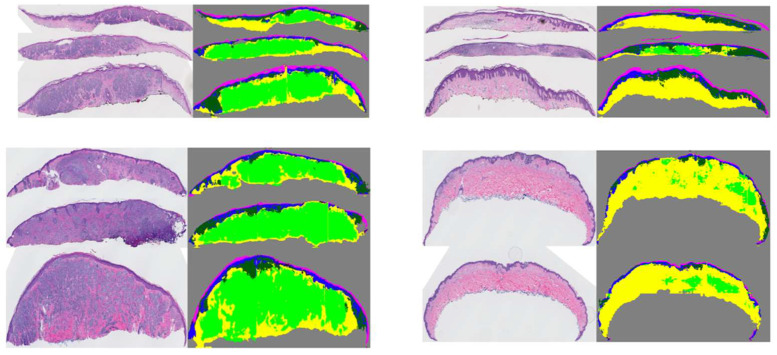
Examples of original WSIs and their corresponding segmentation mask. The segmentation images contain the dermis, epidermis, stratum corneum, background, dermal, and epidermal nests. The model was trained on coarse and sparse annotations.

**Figure 5 diagnostics-12-01713-f005:**
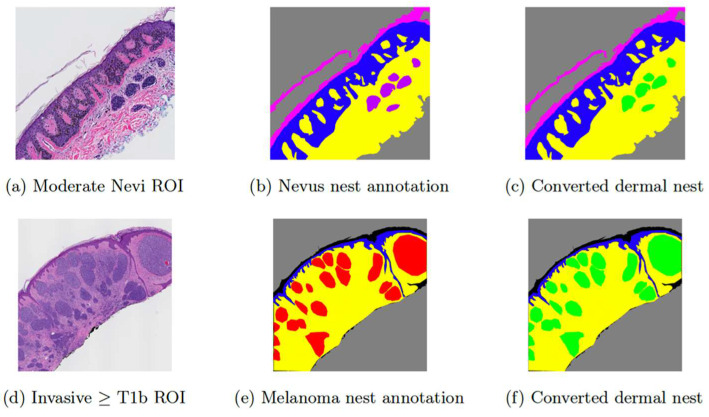
Examples of input ROI images and their corresponding annotations. (**a**) shows a moderate Nevi ROI image, (**b**) shows the original Nevus dermal nest annotation in purple, (**c**) is the converted version of (**b**) in which purple annotations of nevus dermal nests (DMN-N) are converted to green markings of dermal nest (DMN), (**d**) shows an invasive melanoma stage ≥ T1b ROI image, (**e**) shows the original melanoma nests annotation in red, (**f**) is the converted version of (**e**) in which red annotation of melanoma dermal nests (DMN-M) are converted to green markings of dermal nest (DMN).

**Figure 6 diagnostics-12-01713-f006:**
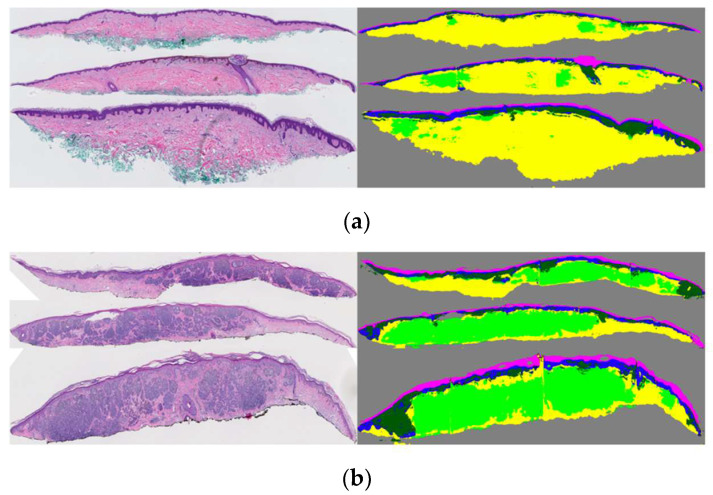
Example of dermal nest segmentation (in light green) on WSI: (**a**) a moderate nevi case; all the dermal nests are nevi type. (**b**) An invasive melanoma stage ≥ T1b case; the segmented dermal nests might contain both nevi and melanoma dermal nests.

**Figure 7 diagnostics-12-01713-f007:**
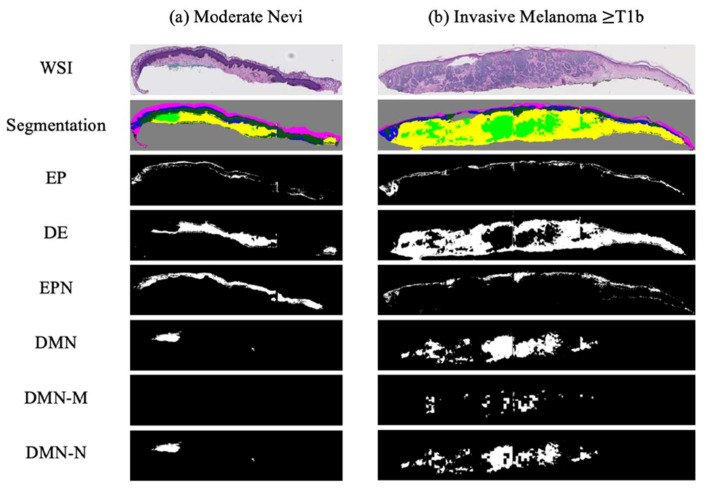
Examples of binarized segmentation masks: (**a**) a moderate nevi case; (**b**) an invasive melanoma stage ≥ T1b. From top to bottom, one extracted slice from a WSI, all segmentation masks in one mask (containing EP, DE, EPN, and DMN), binary Epidermis (EP) mask, binary dermis (DE) mask, binary epidermal nest (EPN) mask, binary dermal nest (DMN) mask, binary melanoma dermal nest (DMN-M), and binary nevus dermal nest (DMN-N) mask are shown.

**Figure 8 diagnostics-12-01713-f008:**
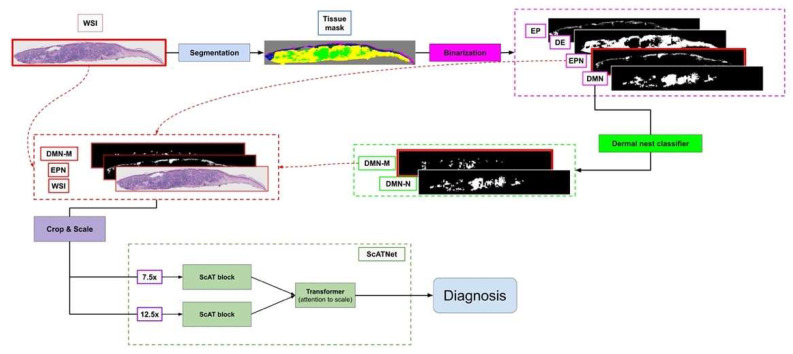
Overview of our diagnosis pipeline. The WSI goes to the segmentation pipeline to generate a tissue segmentation mask. Then, four clinically important tissue structures: epidermis (EP), dermis (DE), epidermal nest (EPN), and dermal nest (DMN) will be extracted into four corresponding binary masks. Extracted Dermal Nests will go through a dermal nests classification step to generate two sub-categories of melanoma dermal nest (DMN-M) and nevus dermal nest (DMN-N). Then, the selected tissue masks based on the experiment will be concatenated to the RGB channels of the WSI image. Each image will be cropped into smaller patches afterward. The patches go through the ScATNet pipeline that extracts patch embeddings, then, using contextualized patch-embedding and scale-aware embedding across available scales, chooses the diagnostic class of the case from mild and moderate dysplastic nevi (MMD), melanoma in situ and severely dysplastic nevi (MIS), invasive melanoma T1a (T1a) and melanoma invasive ≥ T1b (T1b). Note that the concatenated masks to the WSI (DMN-M and EPN) and ScATNet scales (7.5× and 12.5×) shown in this figure are just one example of our multiple experimental studies.

**Figure 9 diagnostics-12-01713-f009:**
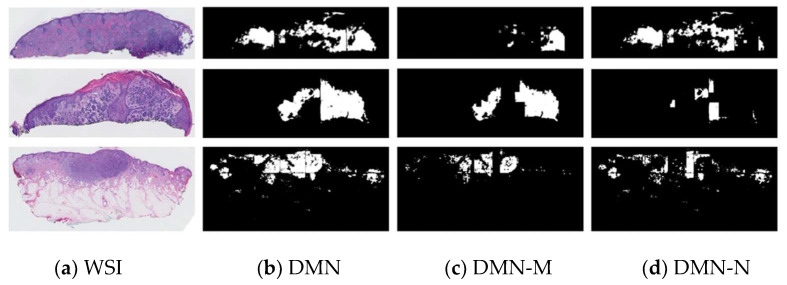
Examples of our best nest classifier, ResNet’s results on WSI: (**a**) extracted slices of invasive melanoma WSIs; (**b**) dermal nest results of segmentation model; (**c**) melanoma dermal nest (DMN-M) portion of DMN; (**d**) nevus dermal nest (DMN-N) portion of DMN.

**Figure 10 diagnostics-12-01713-f010:**
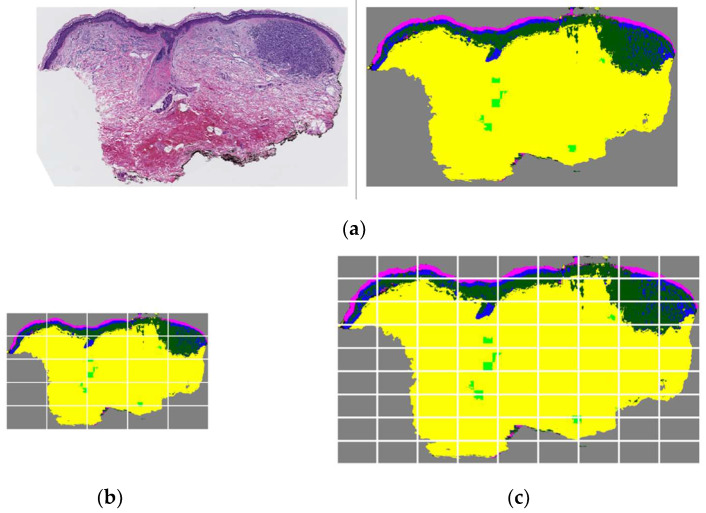
Low-resolution vs. high-resolution patching when there is an inaccurate segmentation mask in the testing case. (**a**) A WSI and corresponding segmentation mask that includes dermis, epidermis, melanoma dermal nest, epidermal nest, corneum, and background. In this example case, epidermal nests are inaccurately segmented and over-labeled. (**b**) The segmentation mask in 7.5× scale divided into 25 crops as input patches for ScATNet. (**c**) The segmentation mask in 12.5× scale divided into 81 crops as input patches for ScATNet. There is a higher number of patches with inaccurate and noisy segmentation on the 12.5× scale compared to the 7.5× scale, which possibly led to a false prediction on the 12.5× scale using ScATNet.

**Table 1 diagnostics-12-01713-t001:** Distribution of diagnostic categories in M-Path data.

Diagnostic Category	Number of Cases
Class I (e.g., Mildly Dysplastic Nevi)	25
Class II (e.g., Moderately Dysplastic Nevi)	36
Class III (e.g., Melanoma in Situ)	60
Class IV (e.g., Invasive Melanoma Stage T1a)	72
Class V (e.g., Invasive Melanoma Stage ≥ T1b)	47
**Total**	**240**

**Table 2 diagnostics-12-01713-t002:** Quantitative nest classification results on ROIs-CNN models.

Method	F-Score	Precision	Sensitivity	Specificity
DenseNet	0.88	0.87	0.89	0.82
ShuffleNet	0.78	0.80	0.76	0.74
ResNet	0.96	0.95	0.97	0.93

**Table 3 diagnostics-12-01713-t003:** Experimental results of WSI diagnosis along with segmentation masks.

Experiments	F-Score *				Sensitivity **	Specificity **
	Average	Min	Max	Median		
**WSI + EPN + DMN-M**	**0.60**	**0.58**	**0.63**	**0.59**	**0.60**	**0.87**
WSI + EPN + DMN	0.57	0.54	0.61	0.56	0.57	0.85
WSI + EPN + DMN-M + DMN-N	0.56	0.53	0.60	0.55	0.56	0.85
WSI + EP + DE + EPN + DMN	0.55	0.53	0.59	0.54	0.55	0.85
WSI	0.54	0.53	0.58	0.54	0.54	0.85
WSI + EPN	0.54	0.52	0.58	0.53	0.54	0.85
WSI + DMN	0.54	0.51	0.56	0.54	0.54	0.85
WSI + DMN-M + DMN-N	0.54	0.52	0.55	0.54	0.54	0.86
WSI + DMN-M	0.52	0.50	0.55	0.51	0.52	0.84

* F-score is reported for 10 random seeds; ** sensitivity and specificity are average scores over 10 random seeds per experiment.

**Table 4 diagnostics-12-01713-t004:** Comparison of two confusion matrices. Rows are defined by expert consensus and columns are by model predictions. (**a**) An example experiment with only WSI and no segmentation mask. (**b**) An example experiment of WSI + EPN + DMN-M.

	MMD	MIS	T1a	T1b			MMD	MIS	T1a	T1b
MMD	**17**	8	4	0		MMD	**17**	9	3	0
MIS	7	**12**	9	2		MIS	3	**16**	10	1
T1a	0	9	**18**	4		T1a	5	2	**18**	4
T1b	0	2	9	**12**		T1b	0	0	8	**15**
(**a**) WSI						(**b**) WSI + EPN + DMN-M				

**Table 5 diagnostics-12-01713-t005:** Comparison of F-score results of raw WSI and tissue experiment (WSI + EPN + DMN-M) on single-scale experiments and multi-scale experiments.

Scale	WSI	WSI + EPN + DMN-M
7.5×	0.54	0.60
12.5×	0.56	0.57
7.5× & 12.5×	0.57	0.56
7.5× & 10× & 12.5×	0.57	0.55

**Table 6 diagnostics-12-01713-t006:** Comparison of class-based F-score, sensitivity, and specificity of 187 US pathologists and our best model (WSI + EPN + DMN-M) on the same testing set.

Class	F-Score		Sensitivity		Specificity	
	Pathologists	Ours	Pathologists	Ours	Pathologists	Ours
**MMD**	0.71	0.67	0.92	0.76	0.76	0.81
**MIS**	0.49	0.50	0.46	0.44	0.85	0.89
**T1a**	0.62	0.57	0.51	0.64	0.95	0.79
**T1b**	0.72	0.67	0.78	0.57	0.97	0.96

**Table 7 diagnostics-12-01713-t007:** Comparison of baseline methods with our best model (WSI + EPN + DMN-M).

Method	F-Score	Sensitivity	Specificity
penultimate-weighted [9]	0.44	0.44	0.81
hypercolumn-weighted [9]	0.43	0.43	0.81
DSMIL [10]	0.50	0.50	0.83
ChikonMIL [11]	0.56	0.56	0.85
**Ours ***	**0.60**	**0.60**	**0.87**

* Our model is the tissue experiment with (WSI + EPN + DMN-M).

## Data Availability

The dataset that has been used in this research is a private dataset. Contact J Elmore if you would like to discuss using the dataset.

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
