# Peer review of "Improving the Diagnosis of Skin Biopsies Using Tissue Segmentation"

_diagnostics, 2022, doi:10.3390/diagnostics12071713_

Round 1
Reviewer 1 Report
Using machine learning is innovative in diagnostic, and segmentation is one approach that is valuable in image analysis. However, For this study, there are a few concerns:
1. There is no detailed description of segmentation. There is no way for me to evaluate the technological soundness.
2. If ROI is sufficient for the diagnosis, why do we need WSI?
3. For Figure 1, what are three examples, are they from different categories?
4. What is the difference in terms of efficiency between the current method compared to existing published tools?
Reviewer 2 Report
Dear Authors and Academic Editor,
I'm glad I was able to 'peer-review' this very interesting manuscript. This is a very innovative original article, in which the authors focus on training an artificial intelligence algorithm applied to the differential diagnosis of lesions (classified according to the M-Path Dx scheme) that start from the mildly dysplastic nevus up to the invasive malignant melanoma. I have no questions or clarifications regarding the material and methods and results section, as they are well written, clear and give the opportunity to understand in detail the various steps of the work. Very interesting, among other things, is the subclassification of dermal melanocytic nests into "benign, snowy" and "malignant, melanomatosis". I just feel like suggesting to the authors to broaden the discussion further, perhaps with these references that I found to be very recent on Pubmed.
Finally, I suggest checking out some typoos.
Well done
Minor comments:
References: Please, the style of references is not suitable for MDPI style. Revised it!
Add the following papers:
Wells A, Patel S, Lee JB, Motaparthi K. Artificial intelligence in dermatopathology: Diagnosis, education, and research. J Cutan Pathol. 2021 Aug;48(8):1061-1068. doi: 10.1111/cup.13954. Epub 2021 Jan 26. PMID: 33421167.
Cazzato G, Colagrande A, Cimmino A, Arezzo F, Loizzi V, Caporusso C, Marangio M, Foti C, Romita P, Lospalluti L, Mazzotta F, Cicco S, Cormio G, Lettini T, Resta L, Vacca A, Ingravallo G. Artificial Intelligence in Dermatopathology: New Insights and Perspectives. Dermatopathology (Basel). 2021 Sep 1;8(3):418-425. doi: 10.3390/dermatopathology8030044. PMID: 34563035; PMCID: PMC8482082.
Chen SB, Novoa RA. Artificial intelligence for dermatopathology: Current trends and the road ahead. Semin Diagn Pathol. 2022 Jul;39(4):298-304. doi: 10.1053/j.semdp.2022.01.003. Epub 2022 Jan 14. PMID: 35065872.
Author Response
Response to Reviewer 2 Comments
Point 1: Please, the style of references is not suitable for MDPI style. Revise it!
Response 1: The reference style has been revised.
Point 2: Add the following papers:
Wells A, Patel S, Lee JB, Motaparthi K. Artificial intelligence in dermatopathology: Diagnosis, education, and research. J Cutan Pathol. 2021 Aug;48(8):1061-1068. doi: 10.1111/cup.13954. Epub 2021 Jan 26. PMID: 33421167.
Cazzato G, Colagrande A, Cimmino A, Arezzo F, Loizzi V, Caporusso C, Marangio M, Foti C, Romita P, Lospalluti L, Mazzotta F, Cicco S, Cormio G, Lettini T, Resta L, Vacca A, Ingravallo G. Artificial Intelligence in Dermatopathology: New Insights and Perspectives. Dermatopathology (Basel). 2021 Sep 1;8(3):418-425. doi: 10.3390/dermatopathology8030044. PMID: 34563035; PMCID: PMC8482082.
Chen SB, Novoa RA. Artificial intelligence for dermatopathology: Current trends and the road ahead. Semin Diagn Pathol. 2022 Jul;39(4):298-304. doi: 10.1053/j.semdp.2022.01.003. Epub 2022 Jan 14. PMID: 35065872.
Response 2: Thank you for providing recent relevant publications. We referenced these articles in the discussion of the revised manuscript.
Round 2
Reviewer 1 Report
The authors have made timely changes regarding my previous comments. However, the authors' rebuttal statement did not satisfy my concerns, which I think is a fundermental issue for this manuscript.
1. In the rebuttal statement, the authors assumes AI-generated ROI is incorrect, wihout providing data to support their argument.
2. The authors already stated ROI is important area of WSI for diagnosis and ROI surveying is alreay on par with WSI anlysis. Therefore, I see no benefit using WSI anlysis.
Round 3
Reviewer 1 Report
In the rebuttal statement, the authors did give some acceptable reasons for doing WSI.
However, the authors were still dancing around without making revisions in the manuscript. So let me be clear at this time.
1. The introduction needs a paragraph that properly justifies why the authors choose WSI over ROIs. I personally think this is the novelty that needs to be crystal clear.
2. The authors were absolutely right that a single ROI is not sufficient for now and the future, but that is not my point. Let's assume in the case of surveying multiple ROIs, would it be more efficient than WSI?
Author Response
Response to Reviewer 1 Comments
In the rebuttal statement, the authors did give some acceptable reasons for doing WSI.
However, the authors were still dancing around without making revisions in the manuscript. So let me be clear at this time.
Point 1: The introduction needs a paragraph that properly justifies why the authors choose WSI over ROIs. I personally think this is the novelty that needs to be crystal clear.
Response 2: We added the following paragraph to the introduction section:
In our prior AI-based diagnosis work in pathology, our studies utilize regions of interest (ROI) rather than the larger whole slide images (WSI) [9,13,15]. There are two main reasons we used the full WSI for the current study. First, as described in [16], Mercan et al., in the effort to find diagnostically relevant ROIs on breast biopsy WSI, 74% of the output probability map overlapped with the actual ROIs from pathologist viewing behavior, while 26% did not. If such early probability maps are utilized for diagnosis tasks, there is a chance that important diagnostic information is missed or misused. The second reason behind our approach using WSI relates to the interpretive process used by pathologists as they view, assess, and interpret WSI of skin biopsies using current published definitions for clinical classification. The pathologists’ clinical process and classification systems vary by tissue type – for breast biopsy cases, a single ROI of an area within a duct might suffice to allow the pathologists to come to a diagnosis. However, the process used by pathologists of reviewing skin biopsy image data and the information within skin biopsies used to determine a diagnosis is different – information on the image from larger structural data in addition to image data within small clusters of cells is important to both rule in and rule out different diagnoses. Thus, for a diagnosis of melanoma and its precursors, reviewing information from the larger WSI is required in current clinical practice by pathologists before they can provide a diagnosis.
Point 2. The authors were absolutely right that a single ROI is not sufficient for now and the future, but that is not my point. Let's assume in the case of surveying multiple ROIs, would it be more efficient than WSI?
Response 2: We added the following paragraph to the discussion section:
As mentioned in Section 2, each WSI in the M-path dataset has an expert-assigned ROI that carries either important diagnostic or prognostic information. However, since 1) experts were limited to one ROI per case, and 2) the diagnosis of some skin biopsies requires review of the full whole slide image, we designed our diagnosis pipeline to utilize full WSIs rather than a single ROI per case. One might wonder if multiple ROIs would be sufficient, in place of the WSI and perhaps more efficient. For example, if multiple ROIs were generated by an AI program for use in the diagnosis, these may actually slow down the diagnosis process if provided to human pathologists who are used to their own way of examining slides. If multiple ROIs were identified by expert pathologists and were provided to a computer program classifier, it would not know which (if any) were more important and thus, looking at the WSI is still the best course for the computer diagnosis for skin biopsy specimens.